# Applying Biotechnology in the Propagation and Further Selection of *Vaccinium uliginosum* × (*V. corymbosum* × *V. angustifolium*) Hybrids

**DOI:** 10.3390/plants10091831

**Published:** 2021-09-03

**Authors:** Anna A. Erst, Aleksey B. Gorbunov, Sergey V. Asbaganov, Maria A. Tomoshevich, Evgeny V. Banaev, Andrey S. Erst

**Affiliations:** 1Central Siberian Botanical Garden of the Siberian Branch of the Russian Academy of Sciences, 630090 Novosibirsk, Russia; gab_2002ru@ngs.ru (A.B.G.); cryonus@mail.ru (S.V.A.); arysa9@mail.ru (M.A.T.); alnus2005@mail.ru (E.V.B.); erst_andrew@yahoo.com (A.S.E.); 2Laboratory of Herbarium, National Research Tomsk State University, 634050 Tomsk, Russia

**Keywords:** interspecific hybrids, berry crops, in vitro culture, ISSR analysis

## Abstract

The most serious problem of intergeneric and interspecific hybridization is related to overcoming the reproductive isolation of different species. We assessed the efficiency of reproduction under in vitro conditions and the ex vitro growth capacity of interspecific hybrids of *Vaccinium uliginosum* × (*V. corymbosum* × *V. angustifolium*). The percentage of seed germination in in vitro culture was 88% for *V. uliginosum*, form No. 8 × (*V. corymbosum* × *V. angustifolium*), SC5-8, while it was 42% for *V. uliginosum*, form No. 8 × (*V. corymbosum* × *V. angustifolium*), ‘Northcountry’. The analysis of mean value showed that the multiplication rate increased and the shoot height decreased as the 2-isopentenyl adenine (2iP) concentration was increased in the nutrient medium of the studied hybrids. The maximum rate was achieved using 15 μM 2iP. A detailed analysis of the hybrids indicated that the hybrid variant reliably affected growth and development indicators. Inter simple sequence repeat analysis demonstrated that all analyzed hybrids inherited DNA fragments of the parent plants in various combinations, confirming their hybrid nature. Thus, the use of in vitro methods for the propagation and further selection of genotypes is demonstrated as being an effective approach for developing interspecific hybrids of *V. uliginosum* × (*V. corymbosum* × *V. angustifolium*).

## 1. Introduction

A promising area of blueberry breeding in Siberia is interspecific hybridization involving the most productive resistant varieties and selected forms of half-highbush blueberry (*Vaccinium corymbosum* × *V. angustifolium*, 2n = 48) and a local species of bog blueberry (*V. uliginosum* L., 2n = 48). However, the most serious problem of intergeneric and interspecific hybridization is related to overcoming the reproductive barrier that separates species. The use of in vitro techniques makes it possible to obtain viable hybrids from seeds that have an underdeveloped embryo and to overcome the negative effect of various factors on seed germination and seedling viability at the initial stages of development. In vitro embryo culture can be used to overcome the barriers of nonbreeding in interspecific hybridization of *Vaccinium* species with different levels of ploidy [1]. Furthermore, when treated with colchicine, in vitro culture allows for amphiploid hybrids to be obtained by crossing species of different ploidy, such as *V. corymbosum* × *V. ashei* [2].

Rousi [3] provided the first evidence that amphidiploidy could be useful in breeding blueberries suitable for less favorable areas of cultivation. The report detailed obtaining viable fertile hybrids of the tetraploids of *V. uliginosum* (section Vaccinium) and *V. corymbosum* L. (section Cyanococcus). Similar interspecific hybrids were obtained in Poland, Russia, and other countries [4,5,6], and in Japan using the wild-growing hexaploid race *V. uliginosum* (2n = 6x = 72), and *V. corymbosum* ‘Bluecrop’ [7]. These researchers found that seeds were easier to obtain when applying *V. uliginosum* as a seed parent in comparison with parental pollen, and unilateral cross incompatibility occurred in reciprocal crosses. The resulting hybrid seeds germinated after pretreatment with gibberellin in in vitro culture or in nurseries after stratification [7]. While developing mononuclear tetrads of microspores into mature binuclear pollen grains, Luzyanina [8] revealed a higher amount of defective pollen from *V. uliginosum*, which determined the low pollen quality. Therefore, this species is best used as a maternal plant. Earlier, it was shown that tetraploid bog blueberry (from a population in Novosibirsk region, Russia) easily crosses with tetraploid high blueberry and half-high blueberry (fruit set was 70.6 and 50.0%, respectively) [9]. In interspecific breeding programs, a relatively large number of hybrids are needed to be able to identify adequate genetic variation in the participating species and to avoid inbreeding depression in advanced generations [10].

This research was carried out to assess the possibility of using in vitro techniques to propagate interspecific hybrids of *V. uliginosum* × (*V. corymbosum* × *V. angustifolium*) since previous studies have not examined the in vitro culture of seeds of these hybrids.

## 2. Results

The seed germination percentage of hybrids in in vitro culture was 42% for *V. uliginosum*, form No. 8 × (*V. corymbosum* × *V. angustifolium*), ‘Northcountry’ (*V.ul.*8 × ‘NC’) and 88% for *V. uliginosum*, form No. 8 × (*V. corymbosum* × *V. angustifolium*), ‘SC 5-8’ (*V.ul.*8 × ‘SC 5-8’) (Table 1 and Figure 1a). In total, we received 19 seedlings of *V.ul.*8 × ‘NC’ and 22 seedlings (variants) of *V.ul.*8 × ‘SC 5-8’. The germination rate of seeds in the soil substrate was 80%, and the survival rate was no more than 2%. A comparison of plants obtained in vitro and sown in soil was not carried out due to the unrepresentative sample of the latter. The experimental scheme is shown in Figure 2. As a result, the research showed that 11 variants (78%) of *V.ul.*8 × ‘NC’ and 7 variants (32%) of *V.ul.*8 × ‘SC 5-8’ were capable of morphogenesis (shoot formation) in in vitro culture. Significant differences in the multiplication rate among the studied hybrid variants were noted at the second passage. As for the hybrid of *V.ul.*8 × ‘NC’, its maximum and minimum multiplication rates were 42.5 shoots/explant for variants No. 4-5 and 1.3 shoots/explant for No. 4-1 (Figure 1b,c and Figure 3a). The maximum multiplication factor of the hybrid of *V.ul.*8 × ‘SC 5-8’ was observed in variant No. 7-12 and amounted to 19 shoots/explant, while the minimum was 2.8 shoots/explant in variant No. 7-7. Callus formation was only observed in variant No. 7-11 (Figure 3b). Further research showed that not all variants of the studied hybrids were capable of long-term in vitro cultivation on Anderson nutrient medium supplemented with 5 µM 2-isopentenyl adenine (2iP). Some variants died, while others formed a callus and were excluded from subsequent experiments.

The analysis of the average multiplication rate values showed that this parameter increased along with the 2iP concentration in the nutrient medium (Figure 4 and Figure 5
*p* ≤ 0.05) for hybrids of *V.ul.*8 × ‘NC’ and *V.ul.*8 × ‘SC 5-8’. A detailed analysis of this parameter for variant No. 6-3 showed no notable differences in the multiplication rate in all tested media modifications. However, other variants showed reliable differences when using 15 µM 2iP. Among the variants of *V.ul.*8 × ‘SC 5-8’, the differences among the experimental variants were demonstrated to be insignificant (*p* ≤ 0.05) for variant No. 7-9. However, significant differences from the control were shown only on media with 15 µM cytokinin for variant No. 7-3; hybrid No. 6-2 was characterized by a considerable increase in the multiplication rate as the 2iP concentration increased. The highest multiplication rate was obtained with the nutrient medium to which 15 µM 2iP was added, which had a multiplication rate of 8.9 shoots/explant for variant No. 1-2 of the *V.ul.*8 × ‘NC’ and 3.4 shoots/explant for variant No. 6-2 of *V.ul.*8 × ‘SC 5-8’. Variant No. 4-6 formed only a conglomerate of shoots in this medium. The factor analysis confirmed that both the 2iP concentration and the variant of hybrid influenced the multiplication rate of the studied hybrids (Table 2).

The mean shoot height of the studied hybrids significantly decreased as the 2iP concentration increased in the nutrient medium (Figure 6 and Figure 7, *p* ≤ 0.05). Variant Nos. 4-2, 4-8, and 5-3 of the *V.ul.*8 × ‘NC’ hybrids had no notable differences between the experimental and the control groups. In the nutrient medium lacking growth regulators, variant Nos. 1-1, 1-2, 1-5, and 2-2 had the highest shoot heights of 72.4, 72.2, 73.4, and 71.4 mm, respectively. A detailed analysis of the shoot height parameter in hybrid variants showed that all variants of the hybrid of *V.ul.*8 × ‘SC 5-8’ had significant differences in only the media with 15 µM 2iP. The highest height was observed for variant No. 6-2 when using nutrition without growth regulators, reaching 43.7 mm. The factor analysis confirmed that the shoot height of the studied hybrids was influenced by both 2iP concentration and variants of hybrids (*p* ≤ 0.05). It should be noted that the combination of factors (concentration × variant) did not significantly affect the height of shoots of *V.ul.*8 × ‘SC 5-8’ at *p* ≤ 0.05 (Table 2).

The results demonstrate the efficiency of using growth regulator-free 1/2 Anderson medium at the rooting stage. Applying this medium made it possible to obtain 100% rooting for all studied hybrid variants (Figure 1D–J). Plant acclimatization in *Sphagnum* moss proved to be effective as well whereby it was possible to obtain 90% viable plants for the hybrid of *V.ul.*8 × ‘NC’ and 87% for *V.ul.*8 × ‘SC 5-8’. Subsequent planting in the greenhouse and then in the introduction site showed that not all variants had high viability. Some of the hybrid variants died and were dropped from our experiments, indirectly indicating the heterogeneity of the obtained material. After two years of cultivation, we evaluated the growth parameters of the obtained hybrids (Table 3 and Figure 8).

Variant No. 1-5 differed in the highest shoot height and leaf area as well as in having a more elongated leaf blade shape. Variant No. 1-1 differed in its small size and compact shape with more rounded leaves. It should be noted that we did not trace the regularity among the plant heights in in vitro and ex vitro culture. For example, in in vitro culture, variant No. 1-1 did not differ from variant No. 1-5 in terms of height.

The hybrid origin of the obtained samples was confirmed using the ISSR technique with two primers (UBC825 and UBC811) (Figure 9). Fragments in the range of 250‒1000 base pairs of nucleotides were amplified with the UBC825 marker and in the range of 300‒900 bp with the UBC811 marker. Both markers distinguished parental genotypes well. Samples 10 and 11 (Figure 9) of *V. uliginosum* were clones of the same form of *V. uliginosum* No. 8, so they generated identical amplicon spectra for each marker with both ISSR markers. All analyzed hybrids inherited DNA fragments of parent plants in various combinations, which confirmed their hybrid nature.

## 3. Discussion

According to our data, the germination rate was 80% when sowing seeds of similar hybrid combinations into a soil substrate, but only a few specimens were viable (no more than 2%). In in vitro culture, seed germination was comparable, but the survival rate was 78.6% for the hybrid of *V.ul.*8 × ‘NC’, while it was 31.8% for *V.ul.*8 × ‘SC 5-8’ hybrid. In vitro technologies made it possible to overcome the negative effect of various factors on seed germination and seedling viability at the initial stages of development, thus allowing the preservation of unique genetic material. Pathirana et al. [1] showed that in vitro cultivation of embryoids allowed postzygotic barriers to be overcome during interspecific hybridization of *Vaccinium* species with different ploidy levels. In addition, treatment with colchicine allowed viable amphiploids of *V. corymbosum* × * V. ashei* hybrids to be obtained in in vitro culture [2].

The choice of nutrient media was based on the fact that the mother plant of the obtained hybrids was *V. uliginosum*. Although there are a number of reports of in vitro propagation of *Vaccinium* representatives [11,12,13,14,15,16], only a few are devoted to *V. uliginosum* micropropagation. Cüce et al. [17] showed that *V. uliginosum* plants (populations from Trabzon, Turkey) were successfully propagated in vitro on Woody Plant medium (WPM) supplemented with 1.0 mg/L zeatin and 0.1 mg/L β-indolebutyric acid (IBA). The multiplication rate was 3.46 ± 0.59 shoots/explant, and the shoot height was 34.98 ± 1.51 mm. Zong et al. [18] successfully propagated *V. uliginosum* (populations from Changbai Mountain, China) on Murashige and Skoog medium (MS) with the addition of 0.5 mg/L zeatin and 0.2 mg/L IBA (multiplication rate of 5.3 shoots/explant). Gu et al. [19] noted that successful shoot regeneration was achieved on Debnath and McRae medium supplemented with 2.75 mg/L 2iP and 0.10 mg/L α-indoleacetic acid (IAA). Weng et al. [20] developed an in vitro propagation system for *V. uliginosum* cv. Zishuijing. The optimal medium was WPM supplemented with 6-benzyladenine (6-BA) 1 mg/L and zeatin 1 mg/L. We previously showed that for the propagation of four varieties of *V. uliginosum*, two-stage cultivation is optimal: 2 weeks on Anderson’s nutrient medium supplemented with 20 µM 2iP, and then on the same medium with 5 µM 2iP. The multiplication rate was from six to seventeen shoots/explant, depending on the variety, and the shoot height was 30–35 mm [21,22,23].

Our study showed that the maximum reproduction factor of the studied hybrids was obtained on Anderson nutrient medium with 15 µM 2iP, producing 8.9 shoots/explant for variant No. 1-2 of the *V.ul.*8 × ‘NC’, and 3.4 shoots/explant for variant No. 6-2 of *V.ul.*8 × ‘SC 5-8’. Generally, the hybrid of *V.ul.*8 × ‘NC’ had greater shoot height for all tested variants of nutrient media compared with *V.ul.*8 × ‘SC 5-8’. The greatest height of 43.7 mm occurred for the hybrid shoots on a nutrient medium without growth regulators for variant No. 6-2 of *V.ul.*8 × ‘SC 5-8’, while the heights of variant Nos. 1-1, 1-2, 1-5, and 2-2 of *V.ul.*8 × ‘NC’ were 72.4, 72.2, 73.4, and 71.4 mm, respectively. Our data indicate that the hybrid variant of *V. uliginosum* × (*V. corymbosum* × *V. angustifolium*) considerably affected the growth and development indices, which should be considered to optimize nutrient media and cultivation conditions further. At the same time, there were data showing the reverse trend. It was demonstrated that the growth medium—and not the genotype and physical conditions of cultivation—primarily affected the shoot height and multiplication coefficient of five genotypes of the genus *Vaccinium* spp. [24].

We previously observed that *V. uliginosum* varieties are characterized by different rooting abilities in in vitro culture [25]. Some varieties demonstrated 100% rooting on hormone-free media, while others required preliminary treatment with auxins. The hybrids studied were characterized by 100% rooting on 1/2 A hormone-free nutrient medium.

Our results confirmed that ISSR markers are efficient tools for the discrimination of F1 hybrids of *V. uliginosum* × (*V. corymbosum* × *V. angustifolium*) in controlled crosses. In addition, ISSR markers allow for the easy, fast, inexpensive, accurate, reliable, and simultaneous detection of polymorphisms at multiple loci in the genome using low quantities of DNA. These properties make the markers useful for the genetic analysis of various plants. ISSRs are found to be effective in diversity analyses of lowbush blueberry [25] and lingonberry [26]. To the best of our knowledge, this research is the first to use ISSR markers for hybrid verification in *V. uliginosum* × (*V. corymbosum* × *V. angustifolium*).

## 4. Materials and Methods

The experimental scheme is shown in Figure 2.

### 4.1. Plant Material

Seven samples of seeds from three-species hybrids were placed in in vitro culture: 2 samples of *V.ul.*8 × ‘SC 5-8’, and 5 samples of *V.ul.*8 × ‘NC’, obtained in 2017 at an experimental site of the Laboratory for Food Plant Introduction of the Central Siberian Botanical Garden of the Siberian Branch of the Russian Academy of Sciences (CSBG SB RAS) (Novosibirsk, Russia). Each hybrid variant (seedling) was given an ordinal number (Table 1). The original parent plants of *V. uliginosum* were collected in 2003 in the Kolyvan’ district of Novosibirsk region (the upper reaches of the Shegarka river), and studied at the CSBG SB RAS’ experimental site. The origins of the paternal forms are as follows: ‘Northcountry’ is an interspecific hybrid of *V. corymbosum* (B6) × (*V. corymbosum* × *V. angustifolium*) (R2P4) obtained in Minnesota, USA, in 1973; SC 5-8 is a spontaneous hybrid of *V. corymbosum* × *V. angustifolium*, from which seeds were collected by T. Paal on Prince Edward Island, Canada, the seedlings were grown in the vicinity of Tartu city (Estonia), and were transferred to CSBG SB RAS in 2006.

### 4.2. In Vitro Propagation of Interspecific Hybrids

Seeds were surface sterilized with 20% Domestos solution for 20 min, followed by three times rinsing in sterile distilled water. Seeds were germinated on 0.6% water agar at 24 ± 1 °C under a 16/8 h photoperiod with lighting of 54 μmol m^−2^s^−1^. After 30 days of cultivation, the seedlings were placed on Anderson nutrient medium [27] supplemented with 5 µM 2iP (Sigma-Aldrich, St Louise, MI, USA), 3% sucrose, and 0.5% agar. Seedlings at the second passage were cut into single-node segments and cultured on the same nutrient medium for another 60 days. Then, the microshoots were transplanted twice and cultivated on the same medium with 40 days interval. To assess the effect of cytokinin P on shoot formation, 5 and 15 µM 2iP were used at the fifth passage. Growth regulator-free Anderson medium was used as a control. The microshoots obtained at the stage of multiplication were cut and planted for rooting on half strength growth regulator-free Anderson medium (1/2 Anderson medium). All nutrient media were adjusted to pH 5.7 with KOH before autoclaving.

### 4.3. Acclimatization of Regenerated Plants

Healthy plants with well-developed shoots and roots were removed individually from the culture vessels, washed gently under running tap water and transferred to plastic pots containing *Sphagnum* moss at 24 ± 1 °C under a 16/8 h photoperiod with lighting of 54 μmol m^−2^s^−1^ for 1.5 months. A high relative humidity (above 80%) was maintained for the first 14 days. The acclimatized plants were planted in a greenhouse in plastic seedling trays with a substrate containing neutral and acidic peat in a 2:1 ratio for further 4 weeks and, finally, they were transferred to the nursery.

### 4.4. DNA Isolation and ISSR Analysis

DNA was isolated of fresh leaves using a modified CTAB method [28]. Polymerase chain reaction (PCR) was performed on a C-1000 amplifier (Bio-Rad, Hercules, CA, USA) in 15 μL volume. The standard reaction contained a single *Taq* buffer, a mixture of 0.2 mM each dNTP, 2.5 mM free Mg^2+^, 1 µM ISSR primer, 1.5 ng/µL total DNA, a unit of HS Taq DNA polymerase (Eurogen, Russia). The amplification was carried out as follows: predenaturation at 94 °C—4 min; 38 cycles denaturation at 94 °C—25 s, annealing at 60 °C—30 s, elongation at 72 °C—2 min; final elongation—10 min at 72 °C, and storage at 4 °C until further use. PCR aliquot containing the amplified fragment was stained with SYBR Green I, and analyzed by electrophoresis in 1.7% agarose gel using 1 × TAE buffer at 4 V cm^−1^ voltage. ISSR markers (Eurogen, Russia) were used: UBC825 (Sequence (5′-3′): (AC)_8_T, Tm: 60 °C), UBC811 (Sequence (5′-3′): (GA)_8_C, Tm: 60 °C). The sizes of the PCR products were compared with the molecular size standard 100+ bp DNA ladder (Eurogen, Russia).

### 4.5. Data Analysis

Measurements were replicated twice using 100 microshoots per experiment. Statistical analysis was carried out in Statistica 6.0 (LSD-test, ANOVA, *p* ≤  0.05) (StatSoft Inc., Tulsa, OK, USA). Data are presented as mean and standard errors (M ± SE). To study the different quality plants and leaves, the following indicators were taken into account: plant height (mm), leaf length (mm), leaf area (mm^2^), leaf elongation coefficient. The processing of images of seeds was carried out using the SIAMS Photolab system for obtaining and processing images (Siams, Russia).

## 5. Conclusions

Thus, the study revealed that the survival rate of seedlings of *V. uliginosum* × (*V. corymbosum* × *V. angustifolium*) hybrids in in vitro culture greatly exceeded their viability in a soil substrate. Genotypic differences in the multiplication rate and shoot height among the studied hybrid variants were identified. We showed that the maximum multiplication rate was achieved when using Anderson nutrient medium containing 15 µM 2iP. The factor analysis confirmed that both the 2iP concentration and the variant of hybrid influenced the multiplication rate and shoot height of the studied hybrids. The efficiency of using 1/2 A hormone-free nutrient medium was evident at the stage of rooting. We did not find any regularities between plant heights in in vitro and ex vitro culture. ISSR demonstrated that all analyzed hybrids inherited DNA fragments of the parent plants in various combinations, which confirmed their hybrid nature. Thus, this study demonstrates that applying in vitro methods for the propagation and further selection of genotypes is an effective approach in programs to obtain interspecific hybrids of *V. uliginosum* × (*V. corymbosum* × *V. angustifolium*). These hybrids may be useful as breeding material for creating new high-quality varieties in the future.

## Figures and Tables

**Figure 1 plants-10-01831-f001:**
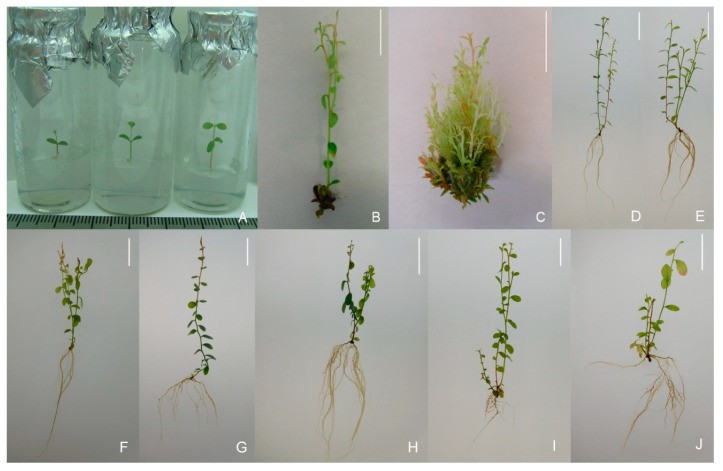
*V. uliginosum* × (*V. corymbosum* × *V. angustifolium*) hybrids in in vitro culture: (**A**) seedlings on 0.6% water agar; (**B**,**C**) microshoots of variant No. 4-1 and variant No. 4-5 on Anderson medium, supplemented 5 µM 2iP; (**D**‒**J**) rooted plants of variant Nos. 1-1, 1-2, 1-5, 2-2, 4-2, 4-8, 7-9 on 1/2 Anderson medium. Bar: 1 cm.

**Figure 2 plants-10-01831-f002:**
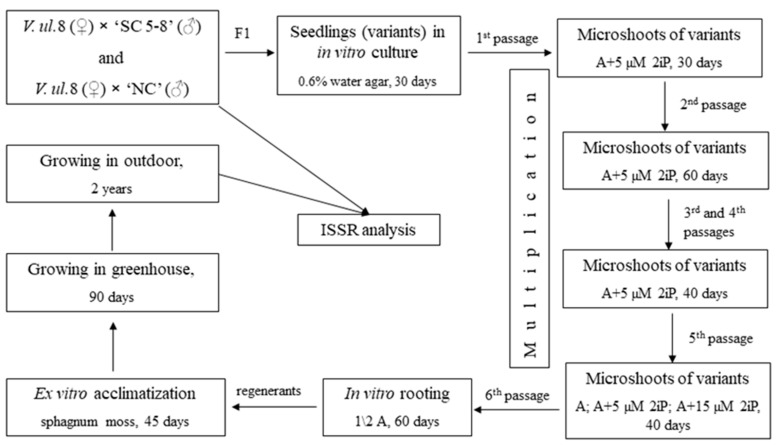
Schematic depiction of the experiment showing stages of in vitro propagation and ex vitro acclimatization, conditions and cultivation periods. Note: *V.ul.*8—*V. uliginosum*, form No. 8; ‘NC’—(*V. corymbosum* × *V. angustifolium*) ‘Northcountry’; ‘SC 5-8’—(*V. corymbosum* × *V. angustifolium*) ‘SC 5-8’; A—Anderson nutrient medium.

**Figure 3 plants-10-01831-f003:**
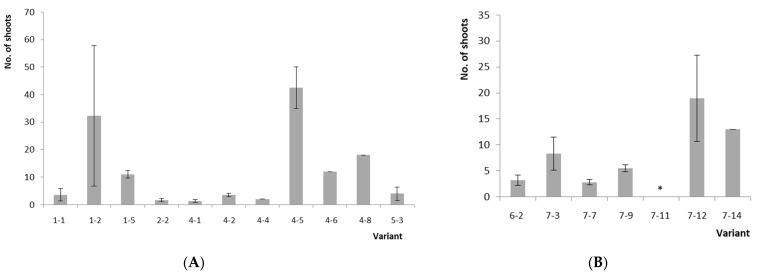
The multiplication rate of variants *V.ul.*8 × ‘NC’ (**A**) and *V.ul.*8 × ‘SC 5-8’ (**B**) on Anderson medium, supplemented 5 µM 2iP at the 2nd passage. Data are presented as mean values with confidence intervals (*p* ≤ 0.05). N = 1–6. Note: *—callusogenesis of variant No. 7-11.

**Figure 4 plants-10-01831-f004:**
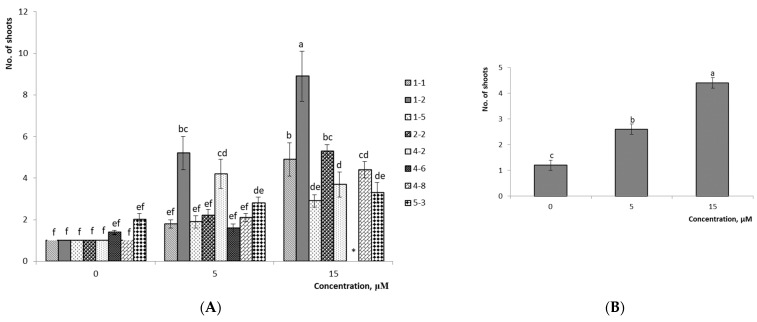
Effect of 2iP concentration on the multiplication rate of *V.ul.*8 × ‘NC’: (**A**) detailed analysis of values of hybrid variants; (**B**) average values of hybrid variants. Means followed by the same letter are not significantly different according to the LSD at *p* ≤ 0.05. Note: *—conglomerate of shoots variant No. 4-6; 1-1, 1-2, 1-5, 2-2, 4-2, 4-6, 4-8, 5-3—variant Nos. 1-1, 1-2, 1-5, 2-2, 4-2, 4-6, 4-8, 5-3.

**Figure 5 plants-10-01831-f005:**
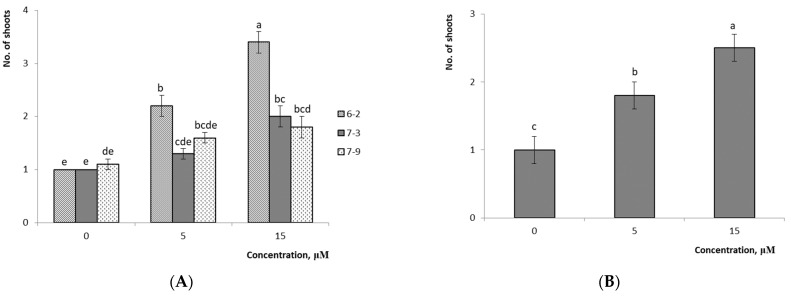
Effect of 2iP concentration on the multiplication rate of *V.ul.*8 x ‘SC 5-8’: (**A**) detailed analysis of values of hybrid variants; (**B**) average values of hybrid variants. Means followed by the same letter are not significantly different according to the LSD at *p* ≤ 0.05. Note: 6-2, 7-3, 7-9—variant Nos. 6-2, 7-3, 7-9.

**Figure 6 plants-10-01831-f006:**
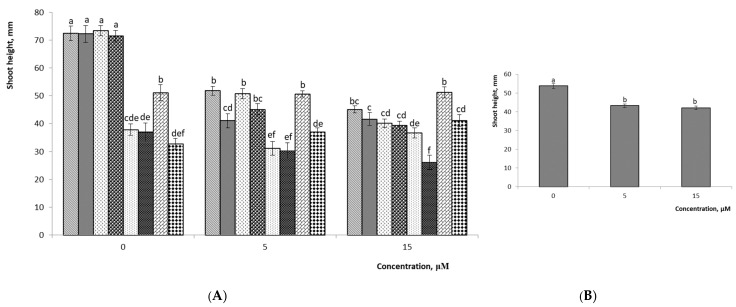
Effect of 2iP concentration on shoot height of *V.ul.*8 × ‘NC’: (**A**) detailed analysis of values of hybrid variants; (**B**) average values of hybrid variants. Means followed by the same letter are not significantly different according to the LSD at *p* ≤ 0.05. Note: 1-1, 1-2, 1-5, 2-2, 4-2, 4-6, 4-8, 5-3—variant Nos. 1-1, 1-2, 1-5, 2-2, 4-2, 4-6, 4-8, 5-3.

**Figure 7 plants-10-01831-f007:**
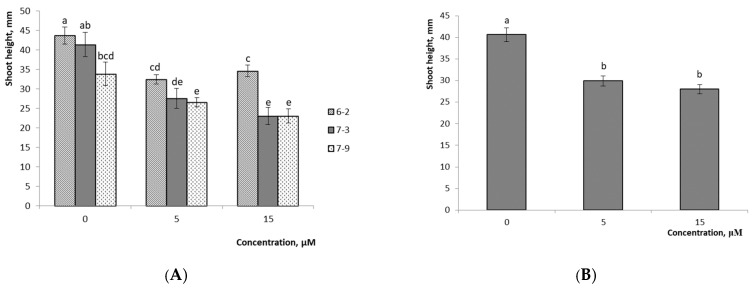
Effect of 2iP concentration on shoot height of *V.ul.*8 × ‘SC 5-8’: (**A**) detailed analysis of values of hybrid variants; (**B**) average values of hybrid variants. Means followed by the same letter are not significantly different according to the LSD at *p* ≤ 0.05. Note: 6-2, 7-3, 7-9—variant Nos. 6-2, 7-3, 7-9.

**Figure 8 plants-10-01831-f008:**
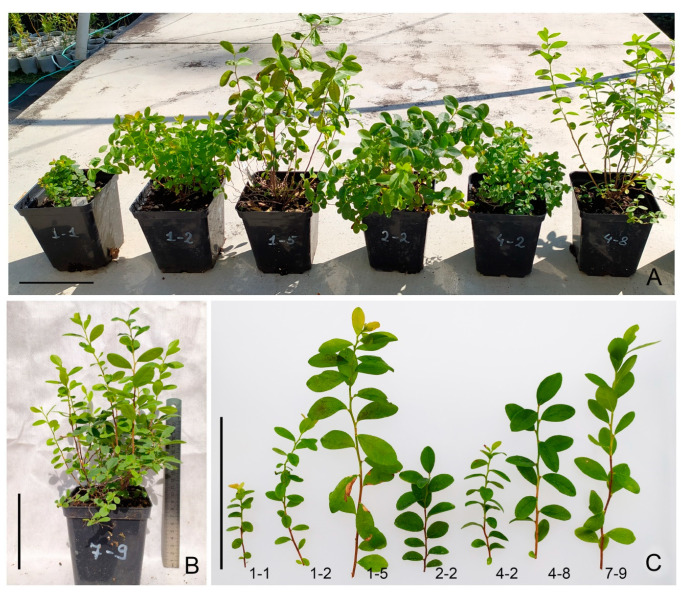
*V. uliginosum* × (*V. corymbosum* × *V. angustifolium*) hybrids after 2 years of cultivation: (**A**) *V.ul.*8 × ‘NC’ No. 1-1, 1-2, 1-5, 2-2, 4-2, 4-8; (**B**) *V.ul.*8 × ‘SC 5-8’ No. 7-9; (**C**) annual shoots of No. 1-1, 1-2, 1-5, 2-2, 4-2, 4-8, 7-9. Bar: 10 cm.

**Figure 9 plants-10-01831-f009:**
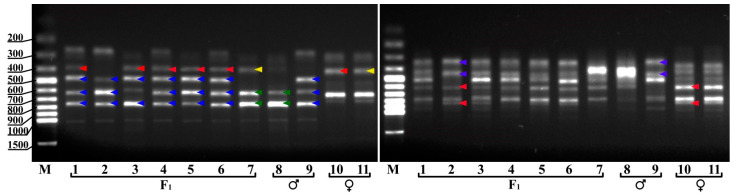
ISSR-PCR eletrophoregrams of genomic DNA with primers UBC825 (**left**) and UBC811 (**right**). 1–6—*V.ul.*8 × ‘NC’ No. 1-1, No. 1-2, No. 1-5, No. 2-2, No. 4-2, No. 4-8; 7—*V.ul.*8 × ‘SC 5-8’ No. 7-9; 8—‘SC 5-8 (♂1)’; 9 -’NC’ (♂2); 10—*V. ul*.8 (♀ 1), 11—*V. ul*. 8 (♀ 2). Note: red arrows—♀ 1-specific markers; yellow arrows—♀ 2-specific markers; green arrows—♂ 1-specific markers; blue arrows—♂ 2-specific markers.

**Table 1 plants-10-01831-t001:** The number of pollinated flowers, set fruits, seeds, and the percentage of germination of *Vaccinium uliginosum* × (*V. corymbosum* × *V. angustifolium*) hybrids on 0.6% water agar in in vitro culture.

Hybrid	Pollination Bag	No. of Pollinated Flowers	No. of Fruits/ Seeds	No. of Seedlings/% of Germination	Variant Abbreviations
*V.ul.*8 × ‘NC’	1	22	3/17	5/29	No. 1-1…No. 1-5
*V.ul.*8 × ‘NC’	2	22	1/8	2/25	No. 2-1, No. 2-2
*V.ul.*8 × ‘NC’	3	11	1/3	0/0	‒
*V.ul.*8 × ‘NC’	4	10	4/17	8/47	No. 4-1…No. 4-8
*V.ul.*8 × ‘NC’	5	6	2/4	4/100	No. 5-1…No. 5-4
*V.ul.*8 × ‘SC 5-8’	6	20	4/6	6/100	No. 6-1…No. 6-6
*V.ul.*8 × ‘SC 5-8’	7	18	5/19	16/84	No. 7-1…No. 7-16

**Table 2 plants-10-01831-t002:** Effect of variant features and concentration of 2iP on the growth and development of *V. uliginosum* × (*V. corymbosum* × *V. angustifolium*) hybrids in in vitro culture (ANOVA).

Effect	*V.ul.*8 × ‘NC’		*V.ul.*8 × ‘SC 5-8’	
	Shoot Height	No. of Shoots	Shoot Height	No. of Shoots
Concentration	*	*	*	*
Variant	*	*	*	*
Concentration × variant	*	*	ns	*

Note: *—significant at *p* ≤ 0.05; ns—not significant.

**Table 3 plants-10-01831-t003:** Growth parameters of *V. uliginosum* × (*V. corymbosum* × *V. angustifolium*) hybrids after 2 years of cultivation.

Variant	Plant Height, mm	Leaf Area, mm^2^	Leaf Length, mm	Leaf Elongation Coefficient
*V.ul.*8 × ‘NC’				
No. 1-1	79.1 ± 13.3 ^d^	49.8 ± 7.3 ^d^	9.7 ± 0.7 ^e^	0.67 ± 0.05 ^a^
No. 1-2	156.2 ± 13.4 ^c^	85.2 ± 17.8 ^d^	15.2 ± 1.9 ^d^	0.52 ± 0.03 ^c^
No. 1-5	250.0 ± 43.0 ^a^	344.1 ± 48.9 ^a^	32.2 ± 2.2 ^a^	0.49 ± 0.04 ^c^
No. 2-2	191.5 ± 44.2 ^b^	244.0 ± 57.1 ^b^	22.4 ± 2.8 ^c^	0.65 ± 0.05 ^ab^
No. 4-2	124.1 ± 19.5 ^c^	133.1 ± 13.8 ^c^	18.5 ± 1.0 ^d^	0.63 ± 0.05 ^b^
No. 4-8	227.2 ± 47.9 ^a^	260.9 ± 57.9 ^b^	26.5 ± 3.3 ^b^	0.52 ± 0.02 ^c^
*V.ul*.8 × ‘SC 5-8’				
No. 7-9	190.3 ± 20.7 ^b^	227.4 ± 47.1 ^b^	25.4 ± 2.2 ^b^	0.52 ± 0.02 ^c^

Note: means followed by the same letter are not significantly different according to the LSD at *p* ≤ 0.05.

## Data Availability

Data available on request due to restriction.

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
