# Peer review of "Applying Biotechnology in the Propagation and Further Selection of Vaccinium uliginosum × (V. corymbosum × V. angustifolium) Hybrids"

_plants, 2021, doi:10.3390/plants10091831_

Round 1
Reviewer 1 Report
Dear authors
I introduced my comments directly on the manuscript.

Author Response
Dear reviewer,
Thank you for your consideration of our manuscript titled ‘Applying in vitro methods for propagation and further selection of Vaccinium uliginosum × (V. corymbosum × V. angustifolium) hybrids’. Your recommendation for improving our team’s manuscript are very important for us and we made all our best performing changes in it. In accordance with your advices we have thoroughly revised the manuscript and consider that correction work is significant. In response to the reviewers’ suggestions we supplemented the manuscript with hybrids growth parameters data in the open field and also corrected all inaccuracies listed by you. We reviewed the manuscript in detail and used the editing services https://www.mdpi.com/authors/english to correct the English language.
Lines 32-34
Point 1: which is inherent to each species. A species is a group of individuals that have reproductive barriers with other species. I suggest to change the sentence, as for instance: "... to overcoming the reproductive barrier that separate species".
Response 1: We corrected the sentence in accordance with your suggestion “However, the most serious problem of intergeneric and interspecific hybridization is related to overcoming the reproductive barrier that separate species.”
Line 34
Point 2: delete “it”
Response 2: we deleted “it”.
Line 64
Point 3: this is not visible in table 1.
Response 3. We corrected the lines in the Table 1. The table 1 has a column “No. of seedlings/% of Germination”. We have replaced the ranges of values “……was 84-100% for V. uliginosum, form No. 8 × (V. corymbosum × V. angustifolium), SC5-8 and 25-100% for V.uliginosum, form No. 8 × (V. corymbosum × V. angustifolium), cv. Northcountry….” with averages values “….was 88%....and….42%....”.
We initiated abbreviations for hybrids: V.ul.8 x 'NC' - V. uliginosum, form No. 8 × (V. corymbosum × V. angustifolium), cv. Norhtcountry; V.ul.8 x 'SC 5-8' - V. uliginosum, form No. 8 × (V. corymbosum × V. angustifolium), SC 5-8.
Line 65
Point 4: Here as in table 1 and figure 2, SC-5-8 comes first. Please see.
Response 4. A detailed description of the parent sample (V. corymbosum × V. angustifolium), SC5-8 is given in the Materials and Methods section. SC5-8 is a spontaneous hybrid of V. corymbosum × V. angustifolium, which seeds collected by T. Paal on Prince Edward Island, Canada. SC5-8 ‒ Seedling from Canada No. 5-8.
Lines 67-68
Point 5: In MM, 4.1 Plat material, there is no mention either to the number of seedlings or to the existance of variants. Why do you consider the 22 seedlings as variants? and what is the definition of variant?
Response 5: The number of viable seedlings is the result of our research, this is not a part of the methodology. The research results are shown in Table 1. The column " No. of seedlings " shows the number of seedlings that we received as a result of crossing (for example 16+6=22). The lines indicate the different parameters for each individual pollination bag. Each seedling is a hybrid variant. We have revised Table 1: we added a columns "Pollination bag" and “Variant abbreviations” and removed the abbreviations 9r.28 and 9r.23. We hope that after the correction, the table has become clearer.
Line 74
Point 6: after the Anderson medium , right? I suggest you to draw a picture with all the steps during in vitro culture, numbering the passages.
Response 6: Yes. It's right. On Anderson nutrient medium supplemented with 5 μM 2iP. We have displayed all stages of cultivation on the scheme (Figure 2. Schematic depiction of the experiments shows stages of in vitro propagation and ex vitro acclimatization, conditions and cultivation periods).
Line 76
Point 7: The identification of the variants is not clear. Why is this 712 and other is 77? How many variants did you get? Variants have to be identified at the begining of "Results".
Response 7: We changed the designation of the variants. For example, the number 7-12 means that this is the variant obtained in the 7th pollination bag, the number after the dash means the seed number (see Table 1).
Line 80
Point 8: ....., respectively.
Response 8: We corrected the sentence “As for the hybrid of V.ul.8 × 'NC', its maximum and minimum multiplication rates were 42.5 shoots/explant for variants No. 4-5 and 1.3 shoots/explant for No. 4-1”.
Table 1
Point 9: Are these the 2 samples referred in MM? Do you distinguish the two samples by the code 9r.28 and 9r.23? This needs to be clarified. The same applies to the cv. Northcountry. In this case, I cannot trace the 5 samples by the code 9r.xx
Response 9: We have revised Table 1: we added a column "Pollination bag" and removed the abbreviations 9r.28 and 9r.23. We hope that after the correction, the table has become clearer.
Lines 99-102
Point 10: the presentation of results should follow an order to facilitate the reading and understanding. In lines 71 and 79, the order of the hybrids was the opposite.
Response 10: we corrected the sentences and figures.
Line 104
Point 11: These corresponds to figure 4, which is presented after figure 3. This text needs revision.
Response 11: we corrected the sentences and figures.
Line 107
Point 12: from the figure, this is not obvious. Could you explain better?
Response 12: for variant 73, there are significant differences only between the control and the culture medium supplemented with 15 μM 2iP. There were no significant differences between the control and the culture medium with 5 μM 2iP and between the culture medium with 15 μM 2iP and 5 μM 2iP.
Line 109
Point 13: this is figure 3.
Response 14: we corrected the sentences and figures.
Line 108
Point 15: there are no results for the 10 uM concentration mention in MM. If you have no results using this intermediate concentration of cytokinin, please remove it from MM or give an explanation for the lack of results.
Response 16: There is an misprint in the materials and methods section. We have not tested the 10 μM concentration.
Line 141
Point 17: This is figure 6 which cannot come before figure 5. Please revise the text.
Response 17: we corrected the sentences and figures.
Line 144
Point 18: have. Please, revise the verbs.
Response 18: we corrected the sentence.
Lines 153, 159
Point 19: or height??
Response 19: we corrected the sentences.
Line 164
Point 20: ir represents only one variant. Please complete the figure with pictures of all variants.
Response 20: we completed the figure with pictures of all variants. We supplemented the article with hybrids growth parameters data in the open field.
Line 166
Point 21: SC-5-8.
Response 21: we corrected the sentence.
Line 168
Point 22: in
Response 22: we corrected the sentence.
Line 169
Point 23: variants
Response 23: we corrected the sentences.
Line 173
Point 24: For instance, variant No. 11...
Response 24: we corrected the sentences.
Line 174
Point 25: delete.
Response 25: we corrected the sentences.
Line 179
Point 26: Please add someting concerning the two mothers!!
Response 26: in the study, two variants of crossing were carried out. For this purpose, two samples of the mother plant were used. Two samples - two vegetatively propagated V. uliginosum, form No. 8.
Lines 181-182
Point 27: This statement needs further discussion. Could you add some discussion taking as an example the band approx. 750 bp (left side)? I suggest you to repeat the electrophoresis using a 3% agarose gel, a high resolution gel or if possible a cappilary electrophoresis (you need to get labelled primers).
Response 27: We have added the arrows mark parent-specific bands.
Line 191
Point 28: When did you get this results? They are not detailed in "Results"
Response 28: we supplemented the research methodology and results with data on the germination and viability of hybrid seeds in the soil.
Line 195
Point 29: SC-5-8
Response 29: we corrected the sentence.
Line 213
Point 30: L
Response 30: we corrected the sentences.
Line 222
Point 31: shoots/explant??
Response 31: we corrected the sentence.
Line 224
Point 32: delete.
Response 32: we corrected the sentence.
Lines 225-228
Point 33: the first part of the sentence does not match with the second part. This needs a revision.
Response 33: we corrected the sentences “Generally, the hybrid of V.ul.8 × 'NC' had greater shoot height for all tested variants of nutrient media compared with V.ul.8 × 'SC 5-8'.
Line 238
Point 34: I cannot understand this statement. How could the genotype be changed by the medium?? Ddi the authors want to mean "phenotype"?
Response 34: we corrected the sentence “It was demonstrated that the growth medium ‒ and not the genotype and physical conditions of cultivation ‒ primarily affected the shoot height and multiplication coefficient of five genotypes of the genus Vaccinium spp.
Lines 245-247
Point 35: How?? Please elaborate more on this considering the band profile and the performance
Response 35: we added a paragraph to the text "Our results confirmed that ISSR markers are efficient tools for the discrimination of F1 hybrids of Vaccinium uliginosum × (V. corymbosum × V. angustifolium) in controlled crosses. In addition, ISSR markers allow for the easy, fast, inexpensive, accurate, reliable, and simultaneous detection of polymorphisms at multiple loci in the genome using low quantities of DNA. These properties make the markers useful for the genetic analysis of various plants. ISSRs are found to be effective in diversity analyses of lowbush blueberry [25] and lingonberry [26]. To our best knowledge, this research is the first to use ISSR markers for hybrid verification in Vaccinium uliginosum × (V. corymbosum × V. angustifolium)."
Lines 250-251
Point 36: seven samples of seeds from three-species hybrids were placed in in vitro culture:
Response 36: we corrected the sentence.
Line 254
Point 37: I suggest to use a , instead "of".
Response 37: we corrected the sentence.
Line 260
Point 38: Shouldn't be V. uliginosum?? This aspect needs to be clearer.
Response 38: The text is correct. The cultivar Norhtcountry is an interspecific 259 hybrid of V. corymbosum (B6) × (V. corymbosum × V. angustifolium) (R2P4).
Lines 262-263
Point 39: ... were grown... , and were transfered... seeds were collected...
Response 39: we corrected the sentences.
Lines 267, 271, 272
Point 40: days
Response 40: we corrected the sentences.
Line 269
Point 41: delete.
Response 41: we corrected the sentence.
Line 274
Point 42: ??? I do not undersand what A is. Is the Anderson nutrient medium?? If yes, then put a A on line 268 where is the first time you mention the Anderson' nutrient medium
Response 42: Yes. It is Anderson nutrient medium. We decided to drop the abbreviation A in the text.
Line 277
Point 43: a fullstop point is missing (.)
Response 43: we corrected the sentence.
Line 283
Point 44: After 1.5 months
Response 44: we corrected the sentence.
Line 287
Point 45: from
Response 45: we corrected the sentence.
Line 289
Point 46: Taq is the name of the bacteria species therefore needs to be written in italic.
Response 46: we corrected the sentence.
Line 290
Point 47: total DNA instead of genomic DNA. When extracting DNA by a CTAB method, mitochondrial and Chloroplast DNA is also extracted.
Response 47: we are agreed with the remark. We have replaced genomic DNA with total DNA.
Line 294
Point 48: I would suggest "amplifed fragments". ISSR always amplifiy more than 1 fragment.
Response 48: we are agreed with the remark. We have replaced the "studied fragment" with "amplifed" fragments.

Reviewer 2 Report
After reading the manuscript, I have ambivalent feelings. It is a quite good example of the application of in vitro cultures in the embryo/seed-rescue techniques to obtain inter-specific Vaccinium hybrids. The Introduction is well written and the aim of the study is properly justified. However, the other parts of manuscripts should be greatly improved. I am not an English language specialist but in my opinion they are written with poor, unclear, and convoluted English. The manuscript has many disadvantages which preclude its acceptance in such a form.
Some other comments:
I think it is appropriate to write the name of cultivar in single quotes (‘Northcountry’) instead “ cv Northcountry “.
I suggest to use abbreviated names of variants/hybrids to simplify/reduce sentences, for example “V.ul.8 x 'NC' “ instead “ V. uliginosum, form No. 8 × (V. corymbosum × V. angustifolium), cv. Norhtcountry “. Give the complete origin of studied variants in M&M.
Use 2iP instead 2-iP.
v278-85 different terms - acclimatization and adaptation. Do use one of them or specific for micropropagation: acclimation, acclimated, etc.
Anderson medium instead A. medium.
Some small errors proving careless preparation of the manuscript, for example: mol m–2s–1 instead mol m–2s–1, mg/L mg/liter (v213-215), Norhtcountry, between instead among, etc. Many grammatical errors; sentences with faulty order. The manuscript should be corrected by a specialist English translator prior to submission
The same data in the text and tables and figures should not be presented.
Materials and Methods
While the only 2 primers were chosen? What were their sequence?
Results
Why the results ‘from greenhouse and the field’ are not presented?
Why the genetic similarity of hybrids, cluster analysis, etc. were not performed? In should be useful in future breeding work.
Table 1. Shifted lines?
Table 2. What is the purpose of placement of ”p≤0.05 p≤0.001 p≤0.05 p≤0.001” in the 2nd line?
Fig 3, 4 The legend not described
Figure 7. Hm, aren’t the pictures with banding patterns placed upside down!? The bands with the highest molecular weight are at the bottom? Add some arrows to mark parent-specific bands.
Discussion
The Authors discussed focused on well known facts, like relationship between cytokinin conc. and shoot multiplication and elongation or clone-specific reaction on the media constituents. The more interesting would be information whether there was any relationship between clone performance in vitro and in vivo? For example – did the clone well proliferating in vitro grow intensively in the greenhouse and the field and was winter-hardy or not? Which primer was better - generated more polymorphic and reproducible banding patterns?
Why the Anderson medium was chosen? Why the only articles devoted to micropropagation of V. uliginosum were cited while the Authors studied hybrids with V. corymbosum × V. angustifolium?
The issue “According to our data, when sowing seeds of similar hybrid combinations into a soil substrate (sand-peat mixture 1:3), the germination rate was 80%, but only a few specimens were viable (no more than 2%). “ (v. 190) is not documented in Results.
Summarizing, despite of some interesting results I recommend to reject the manuscript in such a form. However it may be reconsidered after great improving. I realize that after suggested changes the title of manuscript should be changed , for example for:
Applying in vitro methods and ISSR markers for obtainment? propagation and further selection of Vaccinium uliginosum × (V. corymbosum × V. angustifolium) hybrids
or
Applying biotechnology (techniques) in propagation and further selection of Vaccinium uliginosum × (V. corymbosum × V. angustifolium) hybrids
Author Response
Dear reviewer,
Thank you for your consideration of our manuscript titled ‘Applying in vitro methods for propagation and further selection of Vaccinium uliginosum × (V. corymbosum × V. angustifolium) hybrids’. Your recommendation for improving our manuscript are very important for us and we made all our best performing changes in it. In accordance with your advices we have thoroughly revised the manuscript and consider that correction work is significant. In response to the reviewers’ suggestions we supplemented the manuscript with hybrids growth parameters data in the open field and also corrected all inaccuracies listed by you. We reviewed the manuscript in detail and used the editing services https://www.mdpi.com/authors/english to correct the English language.
Point 1: I think it is appropriate to write the name of cultivar in single quotes (‘Northcountry’) instead “ cv Northcountry “.
Response 1: we replaced cv Northcountry with ‘Northcountry’
Point 2: I suggest to use abbreviated names of variants/hybrids to simplify/reduce sentences, for example “V.ul.8 x 'NC' “ instead “ V. uliginosum, form No. 8 × (V. corymbosum × V. angustifolium), cv. Norhtcountry “. Give the complete origin of studied variants in M&M.
Response 2: We initiated abbreviations for hybrids: V.ul.8 x 'NC' - V. uliginosum, form No. 8 × (V. corymbosum × V. angustifolium), cv. Norhtcountry; V.ul.8 x 'SC 5-8' - V. uliginosum, form No. 8 × (V. corymbosum × V. angustifolium), SC 5-8.
Point 3: Use 2iP instead 2-iP.
Response 3: we used 2iP instead 2-iP
Point 4: v278-85 different terms - acclimatization and adaptation. Do use one of them or specific for micropropagation: acclimation, acclimated, etc.
Response 4: we have changed the term adaptation to acclimatization throughout the text.
Point 5: Anderson medium instead A. medium.
Response 5: we replaced A medium with Anderson medium.
Point 6: Some small errors proving careless preparation of the manuscript, for example: mol m–2s–1 instead mol m–2s–1, mg/L mg/liter (v213-215), Norhtcountry, between instead among, etc. Many grammatical errors; sentences with faulty order. The manuscript should be corrected by a specialist English translator prior to submission
Response 6: we reviewed the manuscript in detail and used the editing services https://www.mdpi.com/authors/english to correct the English language.
Point 7: The same data in the text and tables and figures should not be presented.
Response 7: We revised the manuscript in detail and removed duplicates from the text and tables and figures.
Point 8: Materials and Methods
While the only 2 primers were chosen? What were their sequence?
Response 8: The resulting hybrids are the result of controlled hybridization, so we are confident that two primers are sufficient. We used ISSR analysis to show whether the hybrids are recombinant. The primers used demonstrate this. The sequences are given in the Materials and Methods section - UBC825 [Sequence (5'-3 '): (AC)8T, 296 Tm: 60°C], UBC811 [Sequence (5'-3'): (GA)8C, Tm: 60°C].
Point 9: Results
Why the results ‘from greenhouse and the field’ are not presented?
Response 9: we supplemented the manuscript with growth parameters data from open ground.
Point 10: Why the genetic similarity of hybrids, cluster analysis, etc. were not performed? In should be useful in future breeding work.
Response 10: We used the ISSR method as an additional method to the morphological method. Cluster analysis, determination of genetic distances, etc. was not included in the objectives of our study, since it was only important to carry out express testing of the samples to confirm their hybrid nature. In further breeding work to select plant samples with specific traits and enhance specific traits, we plan to use cluster analysis and determine genetic distances. The purpose of this study is to show the possibilities and prospects of in vitro methods for breeding work, which allows to preserve and quickly multiply the largest possible set of variants of new hybrid combinations.
In further breeding work, we are planning to use more primers to establish the degree of relationship between hybrid offspring and parent plants. In addition, according to the scheme of in vitro reproduction and ex vitro acclimatization developed by us, new hybrids will be obtained from similar crossing combinations. A detailed analysis of morphological traits, growth parameters, yield, resistance to environmental factors, etc. will be carried out.
Point 11: Table 1. Shifted lines?
Response 11: we have corrected the table 1.
Point 12: Table 2. What is the purpose of placement of ”p≤0.05 p≤0.001 p≤0.05 p≤0.001” in the 2nd line?
Response 12: in the table and throughout the text, we left only p≤0.05.
Point 13: Fig 3, 4 The legend not described
Response 13: we have added captions to legends.
Point 14: Figure 7. Hm, aren’t the pictures with banding patterns placed upside down!? The bands with the highest molecular weight are at the bottom? Add some arrows to mark parent-specific bands.
Response 14: Yes. We turned the figure upside down and gave the molecular weight scale from the left in ascending order from top to bottom. We usually present data in this way (Novikova, T.I., Asbaganov, S.V., Ambros, E.V. et al. TDZ-induced axillary shoot proliferation of Rhododendron mucronulatum Turcz and assessment of clonal fidelity using DNA-based markers and flow cytometry. In Vitro Cell.Dev.Biol.-Plant 56, 307–317 (2020). https://doi.org/10.1007/s11627-019-10049-9).
We have added the arrows mark parent-specific bands.
Point 15: Discussion
The Authors discussed focused on well known facts, like relationship between cytokinin conc. and shoot multiplication and elongation or clone-specific reaction on the media constituents. The more interesting would be information whether there was any relationship between clone performance in vitro and in vivo? For example – did the clone well proliferating in vitro grow intensively in the greenhouse and the field and was winter-hardy or not? Which primer was better - generated more polymorphic and reproducible banding patterns?
Response 15: The result of our work is a set of various breeding material with which further long-term breeding work lies ahead. In further breeding work, we plan to use cluster analysis and determination of genetic distances to select plant samples with specific traits and enhance specific traits. The resulting hybrid seeds are the result of controlled hybridization. We believe that the volume of research carried out at this stage of the work is sufficient, and the data obtained are reliable.
Point 16: Why the Anderson medium was chosen? Why the only articles devoted to micropropagation of V. uliginosum were cited while the Authors studied hybrids with V. corymbosum × V. angustifolium?
Response 16: Anderson nutrient medium is the basis for clonal micropropagation of the family Ericaceae. Since the mother plant is bog blueberry, we focused on the features of in vitro propagation of this particular species when selecting nutrient media. The breeding work is based on the enhancement of the traits of bog blueberries, not half-highbush blueberries.
Point 17: The issue “According to our data, when sowing seeds of similar hybrid combinations into a soil substrate (sand-peat mixture 1:3), the germination rate was 80%, but only a few specimens were viable (no more than 2%). “ (v. 190) is not documented in Results.
Response 17: we supplemented the research methodology and results with hybrid seeds germination and viability data in the soil.
Point 18: Summarizing, despite of some interesting results I recommend to reject the manuscript in such a form. However it may be reconsidered after great improving. I realize that after suggested changes the title of manuscript should be changed , for example for:
Applying in vitro methods and ISSR markers for obtainment? propagation and further selection of Vaccinium uliginosum × (V. corymbosum × V. angustifolium) hybrids
or
Applying biotechnology (techniques) in propagation and further selection of Vaccinium uliginosum × (V. corymbosum × V. angustifolium) hybrids
Response 18: We reviewed the manuscript and supplemented it with new data. We agree with the reviewer and have changed the title of the manuscript to ‘Applying biotechnology in the propagation and further selection of Vaccinium uliginosum × (V. corymbosum × V. angustifolium) hybrids’

Round 2
Reviewer 1 Report
Dear authors
Just a detail concerning the name of enzymes which need to be in italic. There are still places where Taq is not in italic format
Reviewer 2 Report
I've found manuscript sufficiently improved and may be published in Plants.